# Tephra data from varved lakes of the Last Glacial-Interglacial Transition: towards a global inventory and better chronologies on the Varved Sediments Database (VARDA)

Anna Beckett[1], Cecile Blanchet[2], Alexander Brauser[2], Rebecca Kearney[2], Celia Martin-Puertas[1], Ian Matthews[1], Konstantin Mittelbach[2], Adrian Palmer[1], Arne Ramisch[2, 3], Achim Brauer[2]

[1]Centre for Quaternary Research, Department of Geography, Royal Holloway University of London, Egham, TW20 0EX, UK

[2]GFZ German Research Center for Geoscience, Section Climate Dynamics and Landscape Evolution, Telegrafenberg, 14473 Potsdam, Germany

[3]Now at: University of Innsbruck, Innrain 52, 6020 Innsbruck, Austria

*Correspondence to:* A. Beckett (anna.beckett.2020@live.rhul.ac.uk)

## Abstract

The Varved Sediments Database (VARDA) was launched in 2020 and aimed to establish a community database for annually-resolved chronological archives with their associated high-resolution proxy records. This resource would support reproducibility through accessible data for the paleoclimate and modelling communities. In this paper, VARDA has been extended by a  dataset of European tephra geochemical data and metadata to enable the synchronisation of varve records during the Last Glacial-Interglacial Transition (LGIT, here defined as 25 ka BP to 8 ka BP; Beckett et al., (2022)). Geochemical data from 49 known individual tephra layers across 19 lake records have been included, with Lago di Grande Monticchio being the single biggest contributor of geochemical data with 28 tephra layers. The Vedde Ash and Laacher See tephra are the most common layers being found in 6 different records and highlights the potential of refining the absolute age estimates for these tephra layers using varve chronologies and for synchronising regional paleoclimate archives. This is the first stage in a 5 year plan funded by the Past Global Changes (PAGES) Data Stewardship Scholarship to incorporate a global dataset of tephra geochemical data in varve records. Further stages of this project will focus on different regions and timescales.

## 1. Introduction

Varved lake sediment records are annually-resolved archives of climatic and environmental change (Brauer, 2004; Zolitschka et al., 2015), with comparable resolution to ice-cores (Rasmussen et al., 2007). The very nature of these records allows for robust chronologies based on annual layer counts, which can be validated by using independent radiometric dating techniques. Furthermore, other lithological and biological proxy data within these archives can be explored at sub-decadal to seasonal scales (Brauer et al., 2008; Zolitschka et al., 2015). Over the last two decades, there has been an increasing focus on (crypto-) tephra in varved sediments. Improved techniques for extracting tephra from sediments with a low shard concentration (e.g. Merkt et al., 1993; Blockley et al., 2005; Walsh et al., 2021) has enabled distal tephra horizons to be detected in varve lake records, enabling the application of tephrochronology to improve varve chronologies (e.g. Stihler et al., 1992; Wulf et al., 2004, 2016; Palmer et al., 2020), the use of varve chronologies to generate more precise ages for tephra layers (e.g. Lane et al., 2015; Dräger et al., 2017; Walsh et al., 2021) and as a synchronisation tool to better understand the time-transgressive nature of rapid environmental and climatic change at regional scales (Tephra lattices) (Lane et al., 2013; Macleod et al., 2014; Wulf et al., 2016).

Tephra horizons detected within varve sediments are often well constrained, undisturbed and can be precisely dated using the varve chronologies (Lane et al., 2013; Palmer et al., 2020; Walsh et al., 2023). However, a key step in developing a tephrochronology requires a link between the tephra horizon in a sediment archive and an eruption of a known age. This stage is normally undertaken using geochemical data which links the tephra to an eruptive centre (Timms et al., 2019). As more tephra horizons have been detected, there have been important community efforts to improve the accessibility of tephra geochemical datasets. Examples include the RESET Database (Bronk Ramsey et al., 2015) and TephraBase (Newton et al., 2007) which both provide geochemical data and metadata related to the sample analysis. VOLCORE (Mahony et al., 2020), is a more recent addition to tephra databases, providing stratigraphic and geographical data on visible tephra layers discovered in ocean drilling projects.

Further to this, there has been a major increase in the number of varve chronologies reported over the past 30 years and even more recently an increase in papers discussing tephra horizons detected in varve records (see Fig. 1). In 2012, the Varve Working Group (VWG) created a database of varved records in .xmsl file format, containing metadata relating to the chronologies of 108 varve lake records, as discussed in Ojala et al., (2012), but this database lacks specific data from proxies and additional chronological control. The recent development of VARDA (Varved Sediments Database 1.0 (Ramisch et al., 2020)) has provided for the first time a global database of varve sites that includes metadata on site locations, duration of the varve record and the associated proxy data. In this paper, we present an extensive dataset of tephra horizons identified in varved records, together with their published geochemical datasets and metadata as an update to VARDA. This dataset focuses on European varve records on VARDA, specifically during the Last Glacial-Interglacial Transition (LGIT) because of the abundance of sequences with tephra reported in this region. We discuss the nature of lake identification, data collection and the range of records now available within the database.

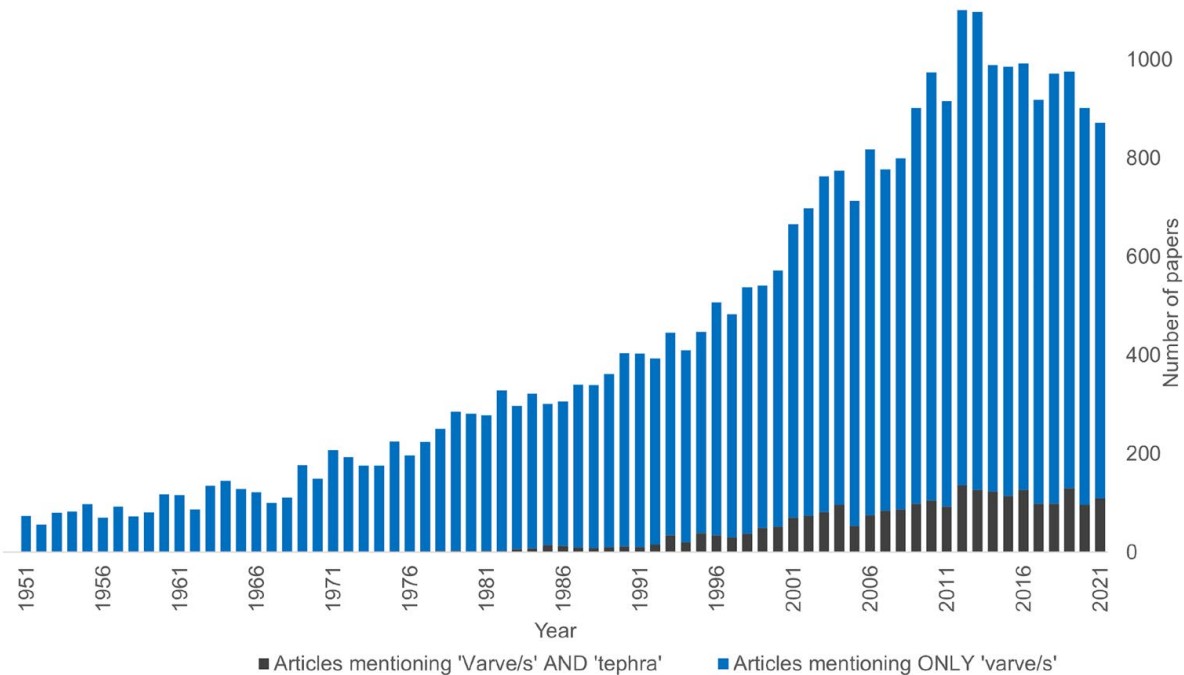

**Figure 1: Results of systematic search of Google Scholar using advanced search functions for each year from 1951 to 2021 using key word searches.**

## 2. Methods
### 2.1. Lake record identification
This work is an initial stage of a five-year programme which aims to reach a global scale and therefore, as a first
step, three criteria were required to be met before tephra data was collected in order to develop the framework for
later stages of the project. Firstly, we defined a region to collect tephra data from. Since the tephrostratigraphies
of different volcanic provinces in Europe are reasonably well developed it was considered that there was sufficient
tephra data to establish the required metadata and the framework could be tested when developing this part of the
database. Secondly, we focused on a specific time period, and, in this case, we chose the LGIT, here defined
broadly between 25 and 8 ka BP. This will enable varved records to be synchronised using tephra during a period
of known abrupt climate change during the last deglaciation. Finally, when tephra layers had been identified within
a published varve record on VARDA, it was essential that those reported tephra layers included tephra
geochemistry and information on the analytical operating conditions including instrument settings and secondary
standards.
Using the pre-existing "age within time span" and "search by continent" features in VARDA (Fig. 2a), lake
records that were within the determined time period and region were narrowed down to a total of 33 records. The
next stage consisted of systematic literature search through the Varve Working Group (VWG) papers and, using
Google Scholar, to identify more recent publications for each lake site and to determine which sites contained
tephra layers.

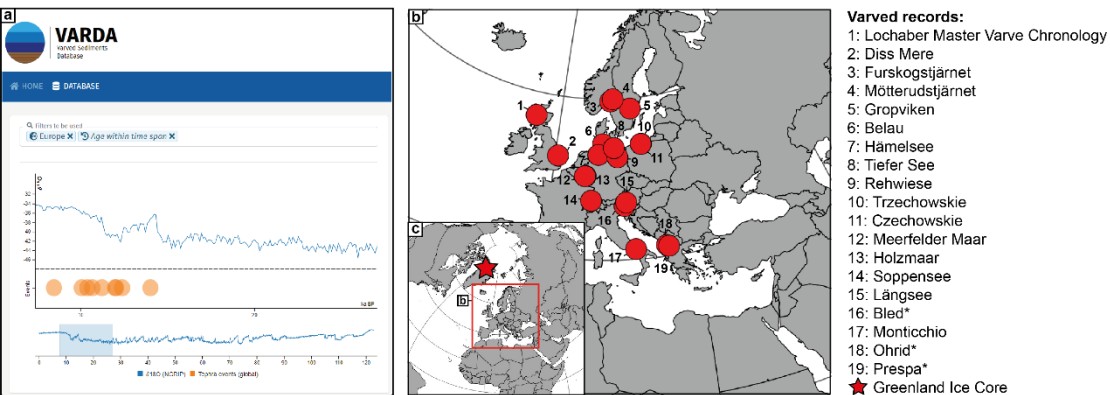

**Figure 2: a)** Screenshot of the parameters used on VARDA to narrow down the search for lakes within the specified time frame and region. (Last accessed: 18/07/2022). **b)** location of all records with tephra geochemical data included in this update. **c)** region where tephra data has been collected, including relative location to the Greenland ice core records. '*' indicates sites that are non-varved.

### 2.2. Data collection

With the aim of adding new proxy-records to VARDA (which is beyond the scope of the present paper), we structured the newly-acquired data using fields identified in Ramisch *et al.* (2020). Where necessary, new fields were adopted in the Beckett et al., (2022) dataset to create a standard approach for documenting and compiling tephra geochemical data in line with established tephra community standards (e.g. Timms *et al.*, 2019; Wallace et al., 2022), and metadata related to the tephra layer as identified by the authors (Table 1 and Table 2). This process generates the relevant information for each individual tephra layer and the sites it has been identified in.

Of the parameters in Table 1, 'Correlation' and 'Source' are mandatory but can be recorded as 'Unknown'. This allows for 1) the input of tephra geochemical data from unknown eruptions and therefore not correlated to a named tephra layer; and 2) allows for the input of tephra layers with an unknown or unconfirmed volcanic source. Tephra layers without a known source or correlation can still be valuable isochronous marker horizons therefore making these fields mandatory was deemed appropriate.

Table 2 outlines all the relevant information published with the geochemical data and provides context to the major element geochemistry. This includes providing age estimates and the methods used for dating each layer, which aids in distinguishing identical geochemical signatures based on age. It must be noted that the 'Age cal BP' provided on the database may vary for the same tephra layer across different sites; defining the 'best' age for a tephra layer is subjective and therefore this project has taken the approach to use the date quoted in the paper publishing the geochemical data. This allows for recalculating ages of the tephra horizon using the most recent $^{14}$C calibration curve, if appropriate. In addition, there has been a recent drive in the tephra community for reporting the analytical conditions used for obtaining geochemistry, and including the standard materials used for calibrating the analytical device. This metadata information enables the data to be reproducible and consistent for future tephra investigations and was therefore collected from the literature for each tephra layer, with future additions to include the published average and two standard deviation measured major and minor element oxide values for secondary standards to ensure quality assurance and accurate tephra correlations.


**Table 1: Mandatory fields for recording tephra geochemical data.**

| Field Name | Field type | Field Description |
| --- | --- | --- |
| **Dataset** | Short text | File name of the original dataset |
| **Lake** | Short text | Name of the lake where the tephra layer was found in |
| **Correlation** | Short text | Name of the correlated tephra layer e.g. Vedde Ash Option for 'Unknown' |
| **Sample ID** | Short text | The lab code of ID used to identify the sample |
| **Source** | Short text | Volcanic origin of the tephra layer Option for 'Unknown' |
| **Lab** | Short text | Laboratory/Institution where analysis was undertaken |
| **Analytical method** | Short text | Type of geochemical analysis undertaken e.g. WDS EPMA |
| **SiO2 wt%** | Number | Weight total % of Silicon (separate fields for raw and normalised values) |
| **TiO2 wt%** | Number | Weight total % of Titanium dioxide (separate fields for raw and normalised values) |
| **Al2O3 wt%** | Number | Weight total % of Aluminium oxide (separate fields for raw and normalised values) |
| **FeO(tot) wt%** | Number | Weight total % of Iron oxides (separate fields for raw and normalised values) |
| **MnO wt%** | Number | Weight total % of Manganese oxide (separate fields for raw and normalised values) |
| **MgO wt%** | Number | Weight total % of Magnesium oxide (separate fields for raw and normalised values) |
| **CaO wt%** | Number | Weight total % of Calcium oxide (separate fields for raw and normalised values) |
| **Na2O wt%** | Number | Weight total % of Sodium oxide (separate fields for raw and normalised values) |
| **K2O wt%** | Number | Weight total % of Potassium oxide (separate fields for raw and normalised values) |
| **P2O5 wt%** | Number | Weight total % of Phosphorus pentoxide (separate fields for raw and normalised values) |
| **SO2 wt%** | Number | Weight total % of Sulphur dioxide (separate fields for raw and normalised values) |
| **Cl wt%** | Number | Weight total % of Chlorine (separate fields for raw and normalised values) |
| **F wt%** | Number | Weight total % of Fluorine (separate fields for raw and normalised values) |
| **Total wt%** | Number | Sum of Weight total % of all elements |



**Table 2: Criteria for meta data relating to individual tephra layers, as identified by the publishing authors.**
**M = Mandatory, O == Optional.**

| Field Name | Field type | Field Description | |
|---|---|---|---|
| **Dated in core** | True/False | Have the publishing authors dated the tephra layers in situ? Either True or False | M |
| **Age transfer reference** | DOI | If previous field False, provide DOI of the reference for the age of the tephra recognised by the authors | O |
| **Age cal BP** | Number | Estimated age of the tephra layer in calibrated years before present (either in situ or external age) | M |
| **Cal age mean** | Number | Mean tephra age (Optional) | O |
| **Cal age median** | Number | Median tephra age (Optional) | O |
| **Uncertainty (+/-)** | Number | Uncertainty of the tephra age in +/- years | O |
| **Sigma** | Number | Confidence window of the age uncertainty: 1 = 68%, 2 = 95.4%, 3 = 99.7%, 4 = 99.9% | O |
| **Calibrated** | True/False | Has the tephra age provided been calibrated in any way? E.g. using 14Cs | M |
| **Calibration curve** | Short text | If "Calibrated = True": calibration curve used for age estimation e.g. IntCal13 | |
| **Dating method** | Short text | Method used for dating the tephra layer e.g. varve counting, 14Cs, age modelling. | M |
| **Depth** | Number | What depth within the lake sequence/core was the tephra identified at? | M |
| **Depth units** | Short text | Unit of measurement for the depth of tephra layers | M |
| **Notes** | Short text | Additional relevant information not aligned with any other field entry | O |
| **Primary data source** | URL | DOI of the primary paper that published the tephra geochemical data | M |
| **Analytical method** | Short text | Method used for obtaining geochemical data e.g. WDS EPMA | M |
| **Analytical instrument** | Short text | Type of analytical instrument used e.g. Cameca SX-100, | M |
| **Beam diameter** | Number | Measured in μm | |
| **Beam current** | Number | Measured in nA | M |
| **Beam Accelerating Voltage** | Number | Measured in kV | M |
| **Secondary Standards** | Short text | Secondary standard material used for measurement of accuracy and precision e.g. Lipari Obsidian | M |



## 3. Results

Of the 33 lakes of suitable age and location on VARDA, 22 contained tephra layers, but only 19 of those have published geochemical data of the tephra layers (locations displayed in Fig. 2b). The lake archives with tephra geochemical data are (Fig. 3, Fig. 4): Belau (Dörfler et al., 2012), Bled (Lane et al., 2011b), Czechowskie (Wulf et al., 2016), Diss Mere (Martin-Puertas et al., 2021; Walsh et al., 2021), Furskogstjärnet (Zillén et al., 2002), Gropviken (Macleod et al., 2014), Hämelsee (Jones et al., 2018), Holzmaar (Wulf et al., 2013), Längsee (Schmidt et al., 2002), Lochaber (Palmer *et al.*, 2020), Meerfelder Maar (Lane et al., 2015), Lago di Grande Monticchio (Wulf et al., 2004, 2008), Mötterudstjärnet (Zillén *et al.*, 2002), Ohrid (Vogel et al., 2010), Prespa (Wagner et al., 2012), Rehwiese (Wulf *et al.*, 2013), Soppensee (Lane et al., 2011a), Tiefer See (Wulf *et al.*, 2016) and Trzechowskie (Wulf *et al.*, 2013). Where applicable, if only part of the lake record fell within the time frame, all tephra layers found in the record, including pre 25 ka BP and/or post 8 ka BP, were compiled to create a consistent approach for each lake record.

Figure 3 displays the interconnections established between the archives through the correlated tephra layers. Within these 19 lake archives, there are 49 individual known tephra layers each with at least one lake archive providing geochemical data. The volcanic source regions for these tephra layers found in Europe are Iceland, Eifel, Massif Central, the Hellenic Arc and Italy, including multiple tephra layers from the Somma-Vesuvius and Campi Flegrei volcanic complexes. There are an additional 24 tephra layers with 'unknown' correlations that have been included in the database. The Vedde Ash (Iceland) and Laacher See Tephra (Eifel) layers are the most commonly found and if combined, allow us to synchronise nine records (Fig. 3). Geographically the Vedde Ash (Katla, Iceland) is the most widespread tephra layer in the database, reaching from Scotland in the West to Sweden and Slovenia in the East (Fig. 4B). Both the Askja-S tephra layer (Askja, Iceland) and Neapolitan Yellow Tuff (Campi Flegrei, Italy) are found in four records across Europe (Fig. 4A and 4D). Lago di Grande Monticchio is the site with the most identified tephra layers at present; there are 28 tephra layers (all originating from Italy or the Hellenic Arc) within the time period of 0 – 100ka BP included in the database but additional layers have been identified earlier in the record (See: Wulf et al., 2012), which will be added to the database in the next steps of the project.

## 4. Implications

The collection of this information is helpful to identify both the temporal (Fig. 3) and spatial range of the tephra layers in predominantly varved (and three non-varved) sediment records across Europe (Fig. 4). Clearly, there is a concentration of tephra layers reported around the Late Glacial period (~15 -11 ka BP) most likely reflecting the wealth of studies focusing on investigating this period of abrupt climate change and its impact on the temperate mid-latitudes of Europe. Nonetheless there is considerable scope to extend these studies to the period immediately after the Last Glacial Maximum in Europe. Recent investigations in mid- and late Holocene tephra layers in European varves show potential for a more robust Holocene tephrostratigraphic framework in the North Atlantic sector (Dräger *et al.*, 2017; Walsh *et al.*, 2021; Walsh *et al.*, 2023). Extending the spatial reach of the tephra database will allow us to build tephra lattices that will help in connecting/synchronising climate records on a global scale.

Comparison of varve records to non-varved records shows where varved sediments with tephra are lacking but
will also provide important information on the potential of finding cryptotephra in varve sequences across Europe
based on the likely passage of the tephra dispersal at the time of the eruption. For an example with comparing to
other well-known tephra databases, Figure 4 displays a kernel density estimation (KDE) of the extent of the Askja-
S, Vedde Ash, Laacher See and Neapolitan Yellow Tuff tephra layers using all known records in the RESET
Database (Bronk Ramsey et al., 2015a) and additional more recent sites that extend the known limit of tephra
dispersal (Wulf et al., 2013; Haflidason et al., 2019; Jones et al., 2020). The KDE in this instance, is used purely
statistically to broadly estimate the 95% confidence interval for spatial distribution of sites containing each tephra
layer (Bronk Ramsey et al., 2015a). Superimposed over this, is a KDE of the tephra dispersal using only the sites
containing these tephra layers in VARDA (Ramisch *et al*., 2020). Furthermore, the location of six additional sites
with varve chronologies (Ammersee (von Grafenstein et al., 1998; von Grafenstein et al., 1999), Gosciaz (Bonk
et al., 2021; Müller et al., 2021), Hancza (Lauterbach et al., 2011b), Lagoon Etoliko (Haenssler et al., 2013),
Mondsee (Lauterbach et al., 2011a; Swierczynski et al., 2013) and Schleinsee (Clark et al., 1989)), which have
high potential for cryptotephra investigations are highlighted (Figure 4). These sites have been identified through
a simple query using VARDA search functions for sites within Europe and within the appropriate time span.

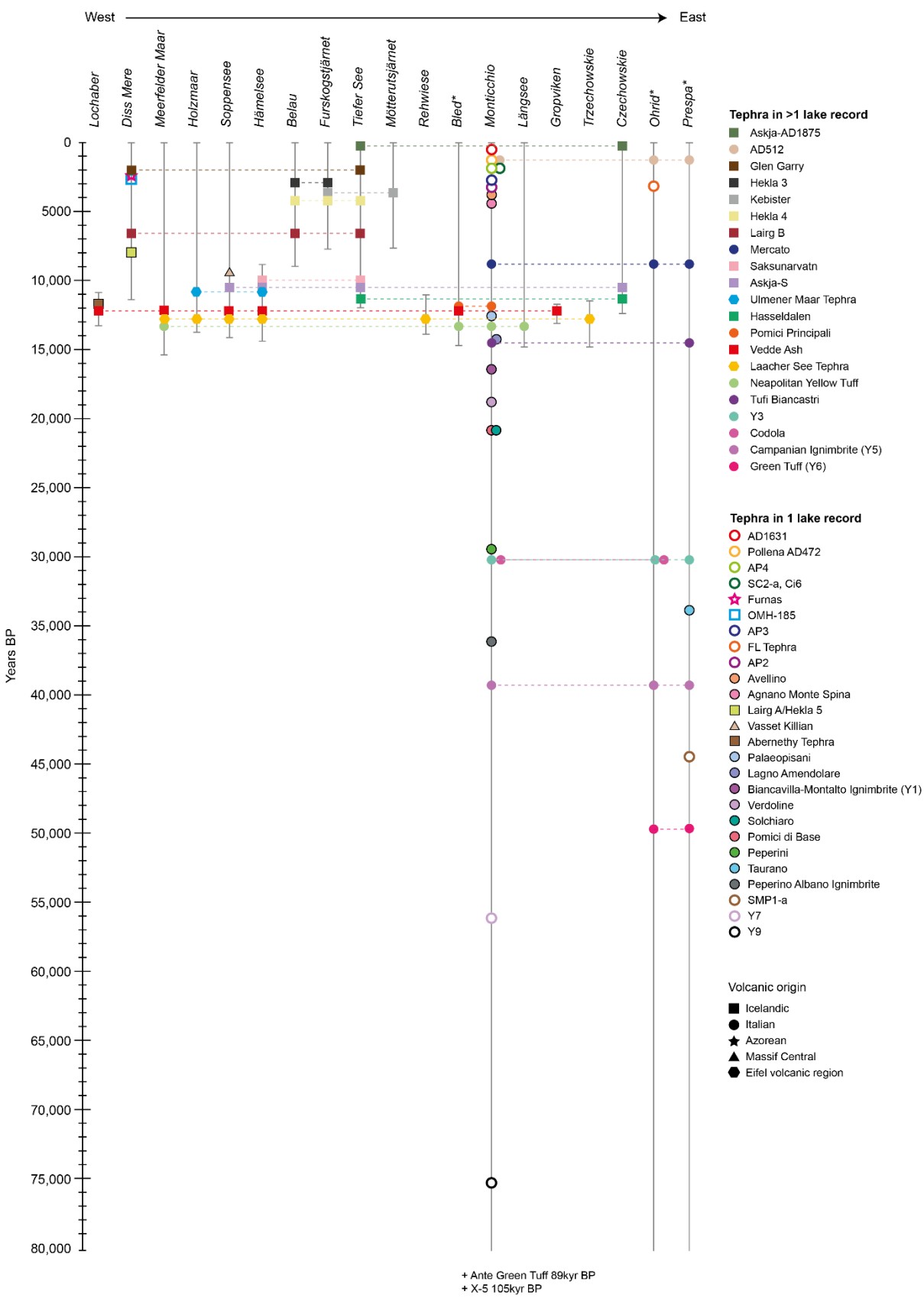

**Figure 3: Connectivity of tephra layers between varved lake records, with dashed lines connecting the same layer between records. Ages used are as detailed in the compiled database. *Records that are non-varved but are included for good chronological control - see: Ramisch *et al*., (2020) for further details.**


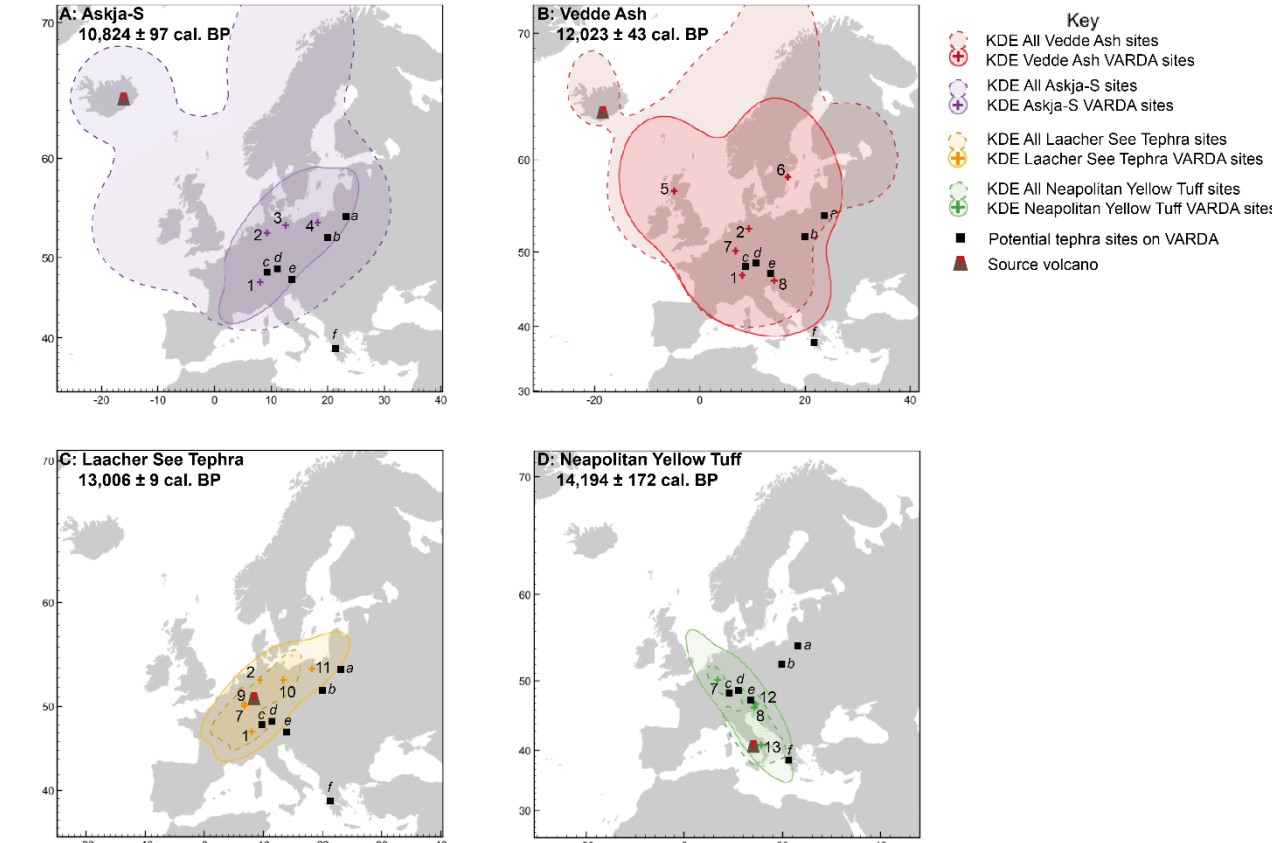

**Figure 4: Kernel Density Estimation plots (Bronk Ramsey *et al.*, 2015a) of four tephra layers present in four or more varve records comparing RESET database supplemented by a selection of more recent identifications that extend the range (dashed line) with the spatial range using the VARDA (solid line). KDE provides a 95% confidence interval on the dispersal range of tephra using the spatial distribution of sites queried. Age estimations sourced from: A) Kearney *et al.*, (2018), B) Bronk Ramsey *et al.*, (2015b), C) Reinig *et al.*, (2021) and D) Bronk Ramsey *et al.*, (2015b). These are the current most precise age estimates for the specific tephra horizons and may not correspond with age estimates in the database.**

**Tephra sites are as follows: 1 Soppensee; 2 Hämelsee; 3 Tiefer See; 4 Czechowskie; 5 Lochaber Master Varve Chronology; 6 Gropviken; 7 Meerfelder Maar; 8 Bled; 9 Holzmaar; 10 Rehwiese; 11 Trzechowskie; 12 Längsee; 13 Lago di Grande Monticchio.**

**Potential tephra sites are: a Hancza; b Gosciaz; c Schleinsee; d Ammersee; e Mondsee; f Lagoon Etoliko.**

**5. Conclusions**
There is much potential in detecting (crypto-) tephra in varved sediment records as they act as one of the most
precise forms of isochronous marker horizons that can help in better understanding the rates of regional climatic
responses to global perturbations. By concentrating on the European tephrostratigraphy during the LGIT, we have
initiated the inclusion of these important datasets, in particular the geochemical information and metadata to
improve accessibility. Further iterations of this expanded database are planned through the PAGES Database
Stewardship Scholarship by extending the spatial coverage and temporal range for tephra horizons in varved
sediments. Expanding the collection of tephra geochemistry provides opportunities to explore novel and emerging
data analysis techniques to identify unknown tephra layers based on their geochemical signatures, potential
dispersal and estimated age. Finally, further research into tephrochronology in varved records should focus on
exploring other regions and time periods with as much intensity as has been given to the LGIT in Europe.

**6. Data availability**
Tephra geochemical data compiled for this project is available open access at the GFZ Data Services
https://doi.org/10.5880/fidgeo.2023.015 (Beckett *et al*., 2022) or via https://varve.gfz-potsdam.de.

**7. Author Contributions**
AnB: Data Curation; Investigation; Validation; Visualisation; Manuscript Writing (original draft &
review/editing). CB: Visualisation; Project administration; Manuscript writing (review/editing). AlB: Database
administration; Data curation; Manuscript writing (review/editing); Software. RK: Manuscript writing
(review/editing); CMP: Conceptualization; Funding acquisition; Manuscript writing (review/editing); Project
Administration. IM: Visualisation; Manuscript Writing (review/editing); KM: Database administration; Software.
AP: Conceptualization; Funding acquisition; Manuscript writing (review/editing); Project Administration;
Supervision. AR: Conceptualization; Project administration. AcB: Manuscript writing (review/editing);
Conceptualization.

**8. Competing interests**
The authors declare that they have no conflict of interest.

**9. Acknowledgements**
We acknowledge the PAGES Data Stewardship Scholarship (No 102) for financial support in the generation and
inclusion of the tephra datasets into the database. This work was supported by the German Federal Ministry of
Education and Research (BMBF) as a Research for Sustainability initiative (FONA; http://www.fona.de, last
access: 10 November 2022) through the Palmod project. Professor Simon Blockley is thanked for his help
accessing and navigating the RESET Database.

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
