# Peer review of "chronologies on the Varved Sediments Database (VARDA)"

_Earth System Science Data, 2023_

## Author Comment (AC1)

**General Responses:**

We thank the three reviewers for their comments, and those who provided community comments too. We will respond to all comments individually but there are some general points mentioned in multiple comments that we would like to address, here labelled as General Responses 1 to 3 (GR-1 - GR-3).

**GR-1**: There is some confusion as to the purpose of this work; this work is an addition to the Varved Sediments Database (VARDA) as opposed to a data compilation exercise using the database. This was not made clear within the text and in the revised manuscript we now explicitly state that this data is a new addition to VARDA (Lines 15, 18, 60, 74).

**GR-2**: The Kernel Density Estimate plots are not meant to be a comprehensive overview of all known findings of the tephra layers, instead they are intended for use as a statistical and schematic diagram to highlight the future potential to better synchronise varve chronologies using tephra layers. We hope that further clarification in the caption of Figure 4 addresses this issue.

**GR-3**: The inclusion of tephra data into VARDA is not intended to be used as a new database for tephrochronologists; we aim that the inclusion of tephra data enables varve chronologists to better synchronise varve chronologies to an absolute timescale using tephra as an isochronous marker horizon.

**Reply to Reviewer #1  (://doi.org/10.5194/essd-2023-154-CC1)**

**We would like to thank Carl Regnéll for the constructive feedback. We copied all comments below, numbered them in order of appearance (CC1-1 to CC1-4) and provided a response accordingly. We hope to have addressed all concerns and improved the manuscript according to the suggestions.**

General comments:

CC1 - 1: It is important not to consider databases like RESET and VARDA as complete and not critically review the data one uses from them, as it might lead to the propagation of misinformation.

Also, when referring to specific sites and studies included in these databases I would consider it only fair to cite the original references for these studies and not only the databases.

More specific:

P.7, Lines 152-154: "Furthermore, the location of seven additional sites with robust varve chronologies, which have high potential for cryptotephra investigations are identified (Figure 4)."

*Authors' response: We appreciate these comments and have adjusted our maps and figures accordingly to remove Lake Aspevatnet from them (CC1-2). In addition we agree that original studies should be referenced in the text for these sites, and regarding the origin of the tephra data for each site we believe we have done this; original site studies and references are all cited on the VARDA database. Where applicable, we have added additional citations (Lines 163 - 166) when referring to sites to reflect this suggestion.*

CC1 - 2: Comments: Potential tephra site "a" (Aspevatnet) is included in the VARDA-database but is not varved, or at least no varves are reported in the reference given in VARDA (Bakke et al. 2005).

*Authors' response: We appreciate this comment as there are a number of sites on VARDA that are not varved; their inclusion into the database is justified in the original VARDA paper (Ramisch et al., 2020) within which we conducted this additional data collection phase for tephra data. The non-varved sites were included originally as they have good chronological control through radiocarbon dating and tephra layers. We have further clarified in diagrams and text where sites are not varved.*

CC1 - 3: Potential tephra site "b" (Storsjom) is misspelt and slightly misplaced on the map. It should be "Storsjön" and as it only has a c. 250 varves long floating chronology (Labuhn et al. 2018) it might not qualify as a "robust varve chronology" with "high potential for cryptotephra

investigations "? In addition, Storsjön was covered by the Scandinavian ice sheet during all of the four eruptions shown in Fig. 4 (e.g. Hughes et al. 2016; Stroeven et al. 2016).

*Authors' response:*  *We thank you for this insight, we have updated Figure 4 accordingly to reflect this.*

CC1 - 4: Fig. 4, p.9: Comment: The known distribution of the Vedde ash is vastly underestimated as it is also found across Arctic Russia and into the Polar Ural Mountains (Haflidasson et al. 2019) and on Svalbard (Farnsworth et al. 2022).

*Authors' response: We appreciate the insight provided here and we acknowledge that the KDE does underestimate the distribution of the Vedde Ash; this is however as a result of the statistical approach used in a KDE that uses 95% confidence. The sites Bolshchoye Shchuchye and Yamozero (Haflidason et al. 2018) were included in the KDE (which has now been acknowledged in the paper) but are at the extreme end of the known extent of the VA and statistically will have been excluded from the 95% interval as a result. The KDE is used here as a schematic representation of the ash dispersal using statistical analysis and is not meant to highlight every location where the VA is found. We have made it clear in the text that additional sites, if not on the RESET Database, have been included in the creation of the KDE maps with citations (Line 159) and further clarified the purpose of the KDE maps in Figure 4 caption.*

---

## Author Comment (AC2)

**General Responses:**

We thank the three reviewers for their comments, and those who provided community comments too. We will respond to all comments individually but there are some general points mentioned in multiple comments that we would like to address, here labelled as General Responses 1 to 3 (GR-1 - GR-3).

**GR-1**: There is some confusion as to the purpose of this work; this work is an addition to the Varved Sediments Database (VARDA) as opposed to a data compilation exercise using the database. This was not made clear within the text and in the revised manuscript we now explicitly state that this data is a new addition to VARDA (Lines 15, 18, 60, 74).

**GR-2**: The Kernel Density Estimate plots are not meant to be a comprehensive overview of all known findings of the tephra layers, instead they are intended for use as a statistical and schematic diagram to highlight the future potential to better synchronise varve chronologies using tephra layers. We hope that further clarification in the caption of Figure 4 addresses this issue.

**GR-3**: The inclusion of tephra data into VARDA is not intended to be used as a new database for tephrochronologists; we aim that the inclusion of tephra data enables varve chronologists to better synchronise varve chronologies to an absolute timescale using tephra as an isochronous marker horizon.

**Reply to Reviewer #2 (://doi.org/10.5194/essd-2023-154-RC1)**

**We are very grateful to Reviewer #2 for the constructive and thorough review that has helped us to improve the manuscript. We copied all comments below, numbered them in order of appearance (RC1-1 to RC1-15) and provided a response accordingly.**

*General comments:*

RC1 - 1: In the present form, it is not clear if this contribution reports on a novel modification/extension of the existing database VARDA or if the presented results are a summary of a VARDA database query. Comparing the structure of VARDA presented in Ramisch et al., 2020 with the actual database website and with the information given about data collection within the manuscripts, some fields (geochemistry) may have been added. However in the manuscripts it is also stated, that this is beyond the scope of this paper. Therefore, I strongly suggest to stress clearly, what is a new or has been modified and what is just a summary of a database query.

***Authors' response:*** *We thank the reviewer for this comment, we have rephrased parts of the manuscript to better reflect the nature of this work as an update to the Varved Sediments Database (Lines 15, 18, 60, 74). Please also see our response to general comment GR-1.*

**RC1 - 2**: It is stated that the project focuses on varved records, but for the integrity of a robust tephra-dataset it is extremely important to consider also data of non-varved records. Important information about the age and glass geochemical composition, which are needed for correct correlations and precise chronologies, may come from such non-varved records and thus would be not considered otherwise. The authors already have partially identified this issue, including data from Lake Ohrid/Prespa, although their non-varved character is not consequently reported (see also community comment, as this seems to be the case for other records). However, the approach stills appears incomplete. As the manuscript is reported to be the initial phase, setting the basis for an overarching project, this needs to be considered in depth for the overall long-term project aim to construct global frameworks.

***Authors' response:*** *We are adding tephra chemistry to the database to serve a chronological purpose (i.e., using the combined strengths of varve records and tephra markers), not to build an additional tephra database. Therefore, we have clarified that this is an update to VARDA in GR-1, which should now provide clarity on this, and we refer back to our response to CC1-2 for inclusion of non-varved sites. An initial phase is used in the sense that we aim to add other varve records that contain tephra from other locations and from an extended chronological range. This is not going to produce the much sought after tephra geochemistry database from a tephrochronologists point-of-view; rather it will help the varve chronologists better understand how to use tephrochronologies.*

**RC1 - 3:** With regard to the compiled dataset of tephra layers, there is no discussion about the quality of the data collected (quality of geochemical analyses, ages). Even if the authors state that the best age of a tephra/eruption may be a subjective feature, the general quality of available ages could be addressed, as in the end for the application of tephrochronology a single age is needed to unify and align chronologies and their records (and it was done in Figure 4).

***Authors' response:*** *We do agree with this comment that data quality should be considered when using tephra layers for the purpose of synchronising records, however, we do not propose in this paper to provide an evaluation of the quality of the data. This is partially a product of no consistent standards on data production through time which limits our ability to provide an assessment of data quality given the standard practices for data publishing have changed through the last few decades. We do provide details of different microprobes used*

*and the operating conditions and a step further would be to include the analytical totals of secondary standards and going forward there is scope to amend this along the lines of community guidelines set out in Wallace et al. (2022). Trace element analysis for MFM is included but as this is the only dataset of this type for the sites identified in the article, it is less of a focus of the paper but is something that will be developed in time as trace element analysis becomes more widely used.*

*Age estimates are quoted from the original paper but do not include any recent updates or remodelled ages that have been included and we make this point clear on line 98. Users querying the VARDA database need to be aware that the ages need to be reconsidered and we suggest that where applicable, ages could be recalibrated using the latest IntCal curve (Line 99-100).*

*Specific comments:*

**RC1 - 4:** *Title:* Consider rephrasing as with the focus set to European volcanism and varved lakes, the dataset collection rather reports on a regional than a global tephra framework. Also, there is no discussion about existing ages, so that chronologies were not improved yet.

**Authors' response:** *As has been addressed in GR-1, we have further clarified that this paper represents an update to VARDA which has an overarching aim to build towards a global inventory of varve records with robust chronologies. We therefore feel with these in text adjustments (and a minor tweak to the title), that the title remains unchanged.*

*Abstract:*

**RC1 - 5:** Consider rephrasing with regard to point out if this contribution represents modification or a query of the VARDA database.

**Authors' response:** *We have taken this on board and updated our abstract and text to make this distinction (Line 14, 18).*

**RC1 - 6:** Please check the given numbers about records and tephra layers. Not all 19 records represent varved records, further Figure 3 shows more than 49 tephra layers…

**Authors' response:** *As has been identified, we are aware that not all sites presented here contain varve sediments, however we now make clear in our figures which records are not varved. Additionally, in the results section (L.117-119), we refer to the sites only as lakes or lake archives and therefore incorporate generally all data collated for the database in this instance. We would like to confirm that there are exactly 49 tephra layers displayed on Figure*

*3, but have clarified in the text that these represent the tephra layers that have been correlated to a known tephra layer, and does not include the tephra layers in the dataset which are uncorrelated (Line 129 and Line 132).*

*Introduction:*

**RC1 - 7:** Figure 1 is not crucially needed with respect to the dataset compiled.

**Authors' response:** *We feel that this figure provides a good visual representation of the increasing interest in combining varve chronologies with tephra layers and explains the wider community need for adding tephra data to structured databases.*

*Methods:*

**RC1 - 8:** It appears that only records registered within VARDA were considered, which may be insufficient to present a full list of (varved) records. Using only data given in VARDA strongly depends on data quality, maintenance and update of this database. This is not discussed within the manuscript. For example, for the Lake Ohrid tephra data the latest results 2019-2023 are not included in VARDA (and were potentially also missed by the google scholar search). Therefore, to provide a reasonable and critical review of existing data to compile a dataset about tephra layers of the LGIT, there should be not only one database considered, but also additional non-database listed references included to ensure completeness and quality of the presented data.

**Authors' response:** *We thank the reviewer for this comment as the Ohrid dataset had indeed been missed. As the database is routinely being updated, this data can be added in the next phase of data compilation. We would further confirm that as this was an addition to the existing VARDA Database, as clarified in GR-1, we are only collating data that would add to the available datasets on VARDA.*

**RC1 - 9:** I would suggest rephrasing of the data collection paragraph in order to point out, how VARDA was modified (see general comment).

**Authors' response:** *We would like to refer here to GR-1 clarifying that this is an addition of a dataset to VARDA.*

**RC1 - 10:** For the new data fields, there are some fields listed in the supplementary data (such as data_availability, datset, lake, geochemistry_availability), which are not given in the tables of the manuscript. There are also only major element data fields (Table 1), whereas also trace element data is given in the attached dataset. For Table 2: What about adding a field for

importing uncalibrated radiocarbon ages in order to simplify recalibration of radiocarbon ages using the same IntCal curve. Please consider adding the information based on which calibration curve the age was calibrated.

*Authors' response: Thank you for this important comment. We agree that there was an inconsistency between the tables in the manuscript and the supplementary data, which we have rectified. Additionally, the intention at the start of the project was to include trace element data but published trace element data was only available for one site, therefore it became less of the focus of the project (but is still valuable information to include). Metadata field "Calibration curve" was added to table 2 in the manuscript and, where possible, to the "Tephra_Major_Elements" sheet in the supplemented dataset, accordingly.*

**RC1 - 11:** Further, for the tables presented I would consider avoiding colour coding of mandatory and optional fields in order to make figures accessible to readers with colour-blindness.

*Authors' response: This is an important point and we replaced the colour coding in table 2 with symbology where "M" stands for mandatory and "O" for optional fields in the manuscript and the published dataset.*

*Results:*

**RC1 - 12:** Referring to the community comment, I also welcome the provided original references of the compiled individual datasets. Please check if your list of references (p.6) is complete and includes latest references (and the consequences for Figure 3).

*Authors response: We appreciate the need to accurately cite original references, and have made these changes accordingly (Lines 159 - 167).*

**RC1 - 13:** Based on the tephra correlations presented in Figure 3, eight of them are reported within the focussed time-interval. Consider if it is applicable to report on the different geochemical results of these eight (everywhere the same composition, variations?) and about their potential ages, findings may also be picked up in the implications section.

**RC1 - 14:** Figure 3: Is it necessary to report tephra layers well beyond this interval? Otherwise, I would suggest highlighting the focussed time-interval.

*Authors response: RC1-13 is is an interesting point raised by reviewer #2, which highlights the utility of the dataset that has been added to VARDA and the potential to explore variation in geochemical composition spatially. However, as we have now clarified the intent of the*

*paper as an update to VARDA, we feel that to properly investigate the spatial differences in tephra composition would require incorporating more geochemical data from non-varved records for a more accurate representation of that tephra layer and we feel that discussing this point would detract from the main focus of the paper. In regard to RC1-14, the addition of tephra layers beyond the time interval is addressed on Line 125 - 127.*

*Implications:*

**RC1 - 15:** I like the idea of comparing the known distribution of an ash cloud with the location of available records to identify potential new targets for (crypto)-tephra investigations. Please specify how the list of (7) new locations was compiled. Maybe it is worthwhile to consider a function for VARDA to report also non-successful cryptotephra investigations, which did not yield any (crypto-)tephra findings. Also these (negative) findings may help to improve knowledge about ash distribution, but also avoid unnecessary investigations by others.

**Authors response:** *We have clarified within the text that the additional potential tephra sites were identified using a simple query on VARDA for sites within Europe and within the appropriate age span (Line 163). We agree that negative findings in a "lessons-learned" database can avoid unnecessary double work for researchers. Negative results are not yet commonly reported in the literature and correct acknowledgement of data ownership for unpublished data and curation of changing those require a more comprehensive data management infrastructure, which is not included at this stage of development.*

---

## Author Comment (AC3)

**General Responses:**

We thank the three reviewers for their comments, and those who provided community comments too. We will respond to all comments individually but there are some general points mentioned in multiple comments that we would like to address, here labelled as General Responses 1 to 3 (GR-1 - GR-3).

**GR-1**: There is some confusion as to the purpose of this work; this work is an addition to the Varved Sediments Database (VARDA) as opposed to a data compilation exercise using the database. This was not made clear within the text and in the revised manuscript we now explicitly state that this data is a new addition to VARDA (Lines 15, 18, 60, 74).

**GR-2**: The Kernel Density Estimate plots are not meant to be a comprehensive overview of all known findings of the tephra layers, instead they are intended for use as a statistical and schematic diagram to highlight the future potential to better synchronise varve chronologies using tephra layers. We hope that further clarification in the caption of Figure 4 addresses this issue.

**GR-3**: The inclusion of tephra data into VARDA is not intended to be used as a new database for tephrochronologists; we aim that the inclusion of tephra data enables varve chronologists to better synchronise varve chronologies to an absolute timescale using tephra as an isochronous marker horizon.

**Reply to Reviewer #3 ([://doi.org/10.5194/essd-2023-154-RC2](://doi.org/10.5194/essd-2023-154-RC2))**

**We appreciate the clear and constructive review by Christine Lane. Many points have been incorporated in our revised version of the manuscript or will be taken on board for future iterations of the dataset (see responses to individual comments below). We copied all comments below, numbered them in order of appearance (RC2-1 to RC2-12) and provided a response accordingly.**

**RC2 - 1**: Beckett et al. report on the addition of information about the occurrence and geochemical compositions of tephra layers within European varve sequences reported in the Varve Database (VARDA). Tephra layers offer the potential to connect varve chronologies at single moments in time with the potential to compare and transfer differential and absolute dating information between sites and increase overall dating precision by replication. The authors state the aim to incorporate information from tephra in varves globally over the coming 5 years, as well as increase the window of time for which data is included. The reasoning for the addition of tephra information is well presented in the manuscript, although as VARDA already includes relevant palaeo proxy datasets for many sites, it is unclear why this particular

"proxy" requires a stand-alone publication. The addition of the "event" layer bar in the GICC05 panel of the VARDA home page is a useful tool for quick reference however, and perhaps as the database grows it will be able to provide a useful online reconnaissance tool for project design and field site selection.

**RC2 - 2**: Whilst this paper addresses the inclusion of tephra data, I found myself browsing VARDA more generally and found myself confused by the inclusion of many non-varved lakes (I'll just note some of the ones I am familiar with as I have worked on them: Lake Victoria, Lake Tanganyika, Lake Bled). I felt a clearer description of the VARDA database itself was probably needed to make sense of the datasets entered to date.

**Authors response:** *The rationale for adding non-varved lakes was originally discussed in Ramisch et al. (2020). We do not seek to modify this rationale in our manuscript but we do acknowledge that a clear reference needs to be made to the original VARDA paper. This clarification has been made in the caption of Figure 3 with reference to the criteria set out in Ramisch et al., (2020) to include non-varved lakes with good chronological control.*

1. Significance

**RC2 - 3**: This particular compilation of tephra layers reported in varve sequences in Europe is unique in that draws together commonalities in the records and could be a great time-saver in looking up sites and articles. Highlighting the value of tephra layers to varve researchers is also beneficial. Additional value could be achieved by including specific and relevant search tools, such as those from the RESET database (Bronk Ramsey et al., 2015) that are used within the article to show the connectivity between records using tephra layers and maps (e.g. KDE) of the sites where tephra layers have been reported. The RESET database is problematic as it is no longer maintained, but in terms of a tephra data repository it is more complete and contains critical data missing from the VARDA database as presented (see comments under Data Quality). The VARDA team might be better to find a means to connect VARDA with that database, rather than starting again to record all of the published tephra layers in Europe within a new repository.

**Authors response:** *We would like to refer here to GR-3 as we do not aim for VARDA to be a new tephra repository but a database that allows varve chronologists to access the available tephra geochemical datasets specifically from varve records.*

**RC2 - 4**: One note that caught my attention was a sentence in the conclusions about the opportunity to explore machine learning approaches to tephra compositional analyses. As there is no mention of this in the body of the paper, it needs further exploration and justification.

If there are additional novel tools being created that could really add to the uniqueness and usefulness of the growing compilation.

**Authors response:** *We highlight this as a potential venture for future work that would be valuable for both the tephra and varve communities, and include it as a suggestion (Line 179).*

2. Data Quality

**RC2 - 5**: Data in VARDA is easily searched and clearly presented and downloadable. I applaud the inclusion of EPMA analytical conditions but I cannot understand the exclusion of secondary standard data, which is critical to evaluating whether one can compare to another tephra dataset or not. The authors referred more than once in the manuscript (e.g. within Table 2 that sets out mandatory and optional criteria for metadata) to the inclusion of "*Standards used for analytical calibration, e.g. Lipari Obsidian*". Two types of standards are used in EPMA work and there seems to be confusion here. Primary standards are usually a suite of minerals or oxides with known elemental compositions, which are used to calibrate the instrument. The publication of primary standard data is not conventional, as what matters is that the data is accurate, not which minerals were used for which elements. Secondary standards are materials of known composition that are analysed before, during and after a run of analyses on an unknown sample, in order to demonstrate the accuracy and precision of the calibration. These are usually matrix-matched to the material being analysed, so the Lipari Obsidian, for example, is a commonly used secondary standard for the analysis of volcanic glass. The tephrochronology community has long called for the inclusion of secondary standard analyses alongside ALL tephra compositional datasets, so that the data may be trusted to make comparisons between tephra datasets generated at different times and on different instruments (e.g. Hunt and Hill., 1996; Kuehn et al., 2011; Wallace et al., 2022). At present, I couldn't find any secondary standard data within the VARDA database, which means that if I were to use it to trace tephra layers, I would immediately have to open the original article and extract the data from there, rather than from VARDA. Those less conscientious might unintentionally propagate poor data and miscorrelations. I strongly recommend that the database authors amend the database to include *either* i. a clear statement that *only* data with secondary standards within 2 standard deviations of published assays are included in the database (a lot of work for data stewards), *or*, ii. secondary standard analyses for all tephra datasets, alongside a link to published assays.

In addition, reviewing the criteria for recording tephra geochemical data, I would recommend that data type (e.g. single grain, whole rock) and material (e.g. glass shards, mineral, pumice) are added as essential criteria. This is also essential metadata for ensuring like is being

compared with like and whilst most data will be single-grain glass shard analyses, it should not be a given.

**Authors response:** *We agree that for accurate comparisons of tephra geochemical data to be made, the secondary standard data needs to be readily available; in this current phase of work, we comment on the need for this data to be included in the next iteration of data to be added to VARDA (L.104). We are aware that secondary standards remain an issue as some information from older papers and projects do not report the analytical totals for secondary standards. We would in the next phase of data collection, aim to include this information which is available for roughly half of the sites mentioned in this article. We will clarify in the text that we are referring to the secondary standards on line 104, which refers to future additions on the database. At present it is our view that users of VARDA should refer to the original papers for secondary standard totals. Future iterations will follow the guidelines set out by Wallace et al. (2022).*

*We renamed the column "standard" to "secondary_standard" in the supplemented data, worksheet "Datasets", and added short references where possible. The columns "secondary_standard_reference_1" and "secondary_standard_reference_2" are added, containing DOI links to the primary references for secondary standards. We added the column "material" to individual samples in the worksheet "Tephra_Major_Elements" and a column "material_description" in the "Datasets" worksheet (supplementary metadata) for a basic classification of the sampled material. We will provide material information in higher granularity in the next iteration of tephra data collection..*

Minor editorial notes

**RC2 - 6**: Title: I would focus here on European and LGIT tephra data in varve records as the potential and value of a global inventory isn't obvious from the article and data at this stage.

**Authors' response:** *We acknowledge that this has been picked up by both reviewers and have made a change to the title, but we feel that clarifying the intent of the paper to provide an update to VARDA helps to solve this issue as it is a global database and this is the first phase of adding to the inventory.*

**RC2 - 7**: Abstract: There is inconsistent use of capitalisation and hyphenation in "last Glacial-Interglacial transition" between the paper title and the abstract that needs correcting one way or the other.

**Authors' response:** *The capitalisation was unified to "Last Glacial-Interglacial Transition" in the title, abstract and text to be consistent with Timms et al. (2019).*

**RC2 - 8**: Introduction, Line 39: the term "well defined" needs explaining.

**Authors' response:** *We agree that the term 'well-defined' is vague in this sentence and have sought to clarify this in the revised article (Line 42) .*

**RC2 - 9**: Methods, Line 73 and 98/99: References to standards used for calibration, rather than secondary standards, needs correcting.

**Authors response:** *This has been corrected in the text (line 76, 104).*

**RC2 - 10**: Figure 2 and 3: Bled, Ohrid and Prespa (at least) are not varved and their inclusion needs an explanation. If non-varved lakes are included, then what do we get from VARDA that is unique? There are asterisks noting that Ohrid and Prespa are non-varved in Fig 3, but not Bled. Other sites I am less familiar with.

**Authors response:** *As previously explained in our response to CC1-2, the inclusion of some non-varved sites is outlined in the original VARDA paper (Ramisch et al., 2020). We do, however, agree that Bled should also be highlighted in this article as not containing varves and have adjusted Figure 2 and 3 to reflect this..*

**RC2 - 11**: Results, lines 126 and 133: The Mediterranean does not describe a volcanic region and it would be better to define to at least Italian and Hellenic Arc, if not specific volcanic fields.

**Authors response:** *We agree with this comment and have made the appropriate changes to reflect a more accurate volcanic origin (Lines 131 and 139).*

**RC2 - 12**: Line 151: I would replace "tephra plume" with "tephra fallout area" to avoid any indication that the sites studied faithfully capture the plume dispersal of an eruption. This is especially pertinent given that within the screen shot of Askja-S sites, Iceland (therefore the volcano) is not included in the shaded envelope.

**Authors response:** *We agree that "tephra plume" may lead to a misinterpretation of the eruption dispersal. According to this suggestion, "tephra plume" was replaced by "tephra dispersal" in 4. Implications (lines 155 and 159).*

---

## Author Comment (AC4)

**General Responses:**

We thank the three reviewers for their comments, and those who provided community comments too. We will respond to all comments individually but there are some general points mentioned in multiple comments that we would like to address, here labelled as General Responses 1 to 3 (GR-1 - GR-3).

**GR-1**: There is some confusion as to the purpose of this work; this work is an addition to the Varved Sediments Database (VARDA) as opposed to a data compilation exercise using the database. This was not made clear within the text and in the revised manuscript we now explicitly state that this data is a new addition to VARDA (Lines 15, 18, 60, 74).

**GR-2**: The Kernel Density Estimate plots are not meant to be a comprehensive overview of all known findings of the tephra layers, instead they are intended for use as a statistical and schematic diagram to highlight the future potential to better synchronise varve chronologies using tephra layers. We hope that further clarification in the caption of Figure 4 addresses this issue.

**GR-3**: The inclusion of tephra data into VARDA is not intended to be used as a new database for tephrochronologists; we aim that the inclusion of tephra data enables varve chronologists to better synchronise varve chronologies to an absolute timescale using tephra as an isochronous marker horizon.

**Reply to Reviewer #4 (://doi.org/10.5194/essd-2023-154-CC2)**

**Our thanks extend to Stefan Wastegård for constructive feedback. All comments were copied below, numbered them in order of appearance (CC2-1 to CC2-4). We hope to have addressed all concerns and improved the manuscript according to the suggestions.**

**CC2 - 1**: I am not familiar with the VARDA database, but I question your decision to include Aspevatnet (not varved) and L. Storsjön (not Storsjom) as potential varve sites for the LGIT. Storsjön was under the Fennoscandian Ice Sheet during most of the LGIT and it would be better to mention south Sweden as a potential area for tephras in glacial varves, as in MacLeod et al (2014) and Devine (2020, PhD thesis).

*Authors response: We appreciate the feedback and agree that these sites would be unsuitable for tephra investigations for this time period. We have adjusted Figure 4 accordingly.*

**CC2 - 2**: Tephra distribution maps should be updated (see also comment by Carl Regnell). I don't fully understand how Kernel maps are constructed, but e.g. Laacher See Tephra has not been found in Latvia, and Vedde Ash and Askja-S not in FInland (not so far, anyway). I think that the maps are misleading and should inlcude all known sites, not only sites in the RESET database.

**Authors response:** *We would refer here to GR-2 (page 1), which explains our intention for the KDE maps as a purely statistical approach to estimating tephra dispersal. We reiterate that the intention of the maps was to compare with the RESET database and the data as it stands on VARDA, but agree that we should have clearly stated in text the up to date known sites that were included in the KDE to begin with. We hope the revised text clarifies this point (Figure 4 caption).*

**CC2 - 3**: Some older references are missing, e.g. Merkt et al. (1993; Boreas) who found Saksunarvatn in four sites in north Germany, of which at least one sequence was varved. This paper could also be cited as an example of improved techniques for cryptotephra extraction along with Blockley et al. (2005) and Walsh et al. (2021). The manuscript has a focus on European deposits, which is understandable, but I think that you should cite the "classic" varve/tephra paper by Stihler et al. (1992; Geology) in the introduction.

**Authors' response:** *The Merkt et al., (1993) paper is a good example of a historic paper which underreports the geochemical data from tephra. In this instance the published dataset uses averages and there is no access to the complete dataset. Consequently, we have excluded sites like this from this new iteration of VARDA. We do however, take on board the acknowledgment of older papers that also improved tephra techniques in the past, and have included these in the introduction (Line 35, 37).*

**CC2 -4** : Small comment: Zillen should be Zillén consistently

**Authors' response:** *We have corrected the spelling of the author name accordingly (lines 120, 123)*

---

## Author Response (AR2)

We thank Christine Lane for her secondary review of the manuscript, we accept we had used the incorrect terminology and have applied the suggestions made which has improved the manuscript.

On behalf of all authors,

Anna Beckett